A preliminary survey of zoantharian endosymbionts shows high genetic variation over small geographic scales on Okinawa-jima Island, Japan

Noda Hatsuko 1
http://orcid.org/0000-0001-8386-3044 Parkinson John Everett 1 2
Yang Sung-Yin 1 3 4
Reimer James Davis 1 5 jreimer@sci.u-ryukyu.ac.jp
1 Molecular Invertebrate Systematics and Ecology Laboratory, Department of Biology, Chemistry and Marine Sciences, Faculty of Science, University of the Ryukyus , Nishihara, Okinawa , Japan
2 Department of Integrative Biology, Oregon State University , Corvallis, OR , USA
3 Microbiology and Biochemistry of Secondary Metabolites Unit, Okinawa Institute of Science and Technology Graduate University , Onna, Okinawa , Japan
4 Biodiversity Research Center , Academia Sinica, Nankang , Taipei, Taiwan
5 Tropical Biosphere Research Center, University of the Ryukyus , Nishihara, Okinawa , Japan
Toonen Robert
Electronic publication date: 2017 Oct 3
Publication date: 2017
Volume: 5
Electronic Location ID: e3740
Received 2017 Jul 11; Accepted 2017 Aug 5
Copyright: © 2017 Noda et al.
Copyright year: 2017
Copyright holder: Noda et al.
License: This is an open access article distributed under the terms of the Creative Commons Attribution License, which permits unrestricted use, distribution, reproduction and adaptation in any medium and for any purpose provided that it is properly attributed. For attribution, the original author(s), title, publication source (PeerJ) and either DOI or URL of the article must be cited.
License URL: https://creativecommons.org/licenses/by/4.0/

Keywords: Diversity, Palythoa tuberculosa, Biogeography, Symbiodinium, Symbiosis

Funding: Japan Society for the Promotion of Science (JSPS) Japan Science and Technology Agency Japan International Cooperation Agency J.E.P was funded by the Japan Society for the Promotion of Science (JSPS). J.D.R was supported by the SATREPS P-CoRIE Project funded by the Japan Science and Technology Agency and the Japan International Cooperation Agency in cooperation with PICRIC and Palau Community College, by a JSPS ‘Zuno-Junkan’ grant entitled ‘Studies on origin and maintenance of marine biodiversity and systematic conservation planning’, as well as by a JSPS ‘Kiban-B’ grant entitled “Global evolution of Brachycnemina and their Symbiodinium”. The funders had no role in study design, data collection and analysis, decision to publish, or preparation of the manuscript.

==============================
Symbiotic dinoflagellates (genus Symbiodinium) shape the responses of their host reef organisms to environmental variability and climate change. To date, the biogeography of Symbiodinium has been investigated primarily through phylogenetic analyses of the ribosomal internal transcribed spacer 2 region. Although the marker can approximate species-level diversity, recent work has demonstrated that faster-evolving genes can resolve otherwise hidden species and population lineages, and that this diversity is often distributed over much finer geographical and environmental scales than previously recognized. Here, we use the noncoding region of the chloroplast psbA gene (psbAncr) to examine genetic diversity among clade C Symbiodinium associating with the common reef zoantharian Palythoa tuberculosa on Okinawa-jima Island, Japan. We identify four closely related Symbiodinium psbAncr lineages including one common generalist and two potential specialists that appear to be associated with particular microhabitats. The sea surface temperature differences that distinguish these habitats are smaller than those usually investigated, suggesting that future biogeographic surveys of Symbiodinium should incorporate fine scale environmental information as well as fine scale molecular data to accurately determine species diversity and their distributions.

Introduction

Symbiodinium is an important genus of dinoflagellate photosymbionts found in tropical and subtropical marine ecosystems. These “zooxanthellae” transfer energy to their invertebrate hosts in nutrient-poor environments, enhancing the growth of reef-building corals and other reef organisms such as zoantharians (Muscatine & Cernichiari, 1969; Baker, 2003). They also serve a key role in establishing the thermal tolerance of coral colonies and shaping the adaptive response of reef organisms to climate change (Sampayo et al., 2008; Thornhill et al., 2014). As climate change intensifies and diversity patterns are expected to alter, it is increasingly important to map marine species distributions on scales both large (McClanahan et al., 2014; Reimer et al., 2017) and small (Mieszkowska & Lundquist, 2011).

The biogeography of Symbiodinium is determined by many factors, including host species distributions and environmental parameters (LaJeunesse et al., 2004). To determine Symbiodinium biogeography, it is critical to use DNA markers with the resolving power to delineate between and among species (LaJeunesse & Thornhill, 2011). Our ability to discern different lineages of Symbiodinium has improved in tandem with the resolution of the molecular markers used to identify them. Originally, highly divergent lineages or “clades” designated A–I were confirmed via the sequencing of 18S and 28S ribosomal DNA (Rowan & Powers, 1991; Pochon & Gates, 2010), followed by the delineation of numerous subclades via phylogenetic analyses of the internal transcribed spacer regions (e.g., internal transcribed spacer 2 (ITS2); LaJeunesse, 2001). More recently, sequences from domain V of the chloroplast large subunit (cp23S) ribosomal DNA (Santos, Gutierrez-Rodriguez & Coffroth, 2003; Kirk et al., 2009; LaJeunesse & Thornhill, 2011; LaJeunesse, Parkinson & Reimer, 2012) and the non-coding region of the plastid minicircle (psbAncr; Takishita et al., 2003; LaJeunesse & Thornhill, 2011) have provided even greater resolution, often corresponding to the species level (LaJeunesse, Parkinson & Reimer, 2012). Microsatellite markers have also been developed to investigate fine-scale diversity in Symbiodinium (Pettay & LaJeunesse, 2007; Grupstra et al., 2017).

As a result of the development and implementation of finer resolution molecular markers, it is apparent that the extent of Symbiodinium diversity may be much greater than previously recognized based on studies using only internal transcribed spacer regions. For example, in examinations of Symbiodinium within the common reef zoantharian Palythoa tuberculosa (Esper, 1805) across a latitudinal gradient of 800 km in the Northern Red Sea, psbAncr sequences revealed up to four unique lineages, despite ITS2 results showing only one single “subclade” lineage (Reimer et al., 2017). The distribution of these individual psbAncr lineages strongly correlated with sea surface temperature (SST) differences of approximately 1 °C. Such results demonstrate a need to re-examine previously reported Symbiodinium diversity with high-resolution markers, as fine scale, ecologically important differences may have been missed.

In biogeography, isolation by distance (IBD) generally refers to how genetic differences between individuals or populations increase with increasing geographical distance due to limitations on dispersal. In Symbiodinium species, other factors in addition to IBD that may act as drivers of evolution include host species associations (LaJeunesse et al., 2004), UV light levels (correlates of turbidity or chlorophyll-a (chl-a) concentrations; LaJeunesse et al., 2010; Tonk et al., 2014), and ocean temperatures (LaJeunesse et al., 2010; Tonk et al., 2014). Correspondingly, differences in Symbiodinium associations have been noted over large oceanic or latitudinal gradients (LaJeunesse & Trench, 2000; LaJeunesse et al., 2004; Reimer et al., 2017), within different host species (Frade et al., 2008a; Thornhill et al., 2014), at different light levels (Frade et al., 2008b; Sampayo et al., 2008; Kamezaki et al., 2013), and in areas of extremely high (Hume et al., 2013) or low ocean temperatures (Chen et al., 2003).

Okinawa-jima Island is ∼1,200 km2 in area, just over 100 km in length, and 3–30 km in width. Although it is not “large” when compared to oceanic or regional scales, the surrounding marine environment encompasses a variety of ecosystems including large bays, muddy tidal flats, patch reefs within lagoons, and fringing reefs (Fujita et al., 2015). Additionally, recent population genetic studies on scleractinian corals (Shinzato et al., 2015), sea cucumbers (Soliman, Fernandez-Silva & Reimer, 2016), and amphipods (White, Reimer & Lorion, 2016) have described unexpected genetic structure among locations around Okinawa-jima Island, particularly between the Kuroshio-influenced west coast and the more isolated east coast.

Based on ITS2 sequence analyses of Symbiodinium within P. tuberculosa in Southern Japan (including Okinawa), we previously reported that subclade C1 or closely related types are dominant (Reimer, Takishita & Maruyama, 2006). Here we use psbAncr sequences to re-examine how much variation exists within Symbiodinium C1 associating with P. tuberculosa, specifically focusing on Okinawa-jima Island’s shallow coral reef environments. We additionally explore potential associations between observed diversity and the environment. This study is intended to complement our recent work in the Red Sea (Reimer et al., 2017) by focusing on the same host species and describing the extent of symbiont diversity on a much smaller geographical and environmental scale.

Materials and Methods

Environmental data

Satellite-derived SST and chl-a data for the waters around Okinawa-jima Island and nearby Amami Oshima Island were acquired from National Aeronautic and Space Administration Giovanni database (https://giovanni.gsfc.nasa.gov/giovanni/; Acker & Leptoukh, 2007), developed and maintained by the NASA Goddard Earth Sciences Data and Information Services Center. Error ranges of the moderate resolution imaging spectroradiometer (MODIS) Aqua data were approximately ±0.25 °C for SST, and ±40% for chl-a. Yearly average SST (SSTavg) and chl-a data and maps used in this study were derived from 4 km resolution data from the MODIS Aqua database. These generated maps provided the basis for estimating annual average SST and chl-a at each sampling location. Data from February 2000 to May 2015 were used for SSTavg analyses, and from July 2002 to May 2015 for chl-a. To examine yearly winter minimum SST (SSTmin) and summer maximum SSTs (SSTmax), we averaged monthly data from February and from August, respectively (2000–2014, n = 14 each).

Specimen collection

The zooxanthellate cnidarian species P. tuberculosa is the most common zoantharian on coral reefs surrounding Okinawa-jima Island (Irei, Nozawa & Reimer, 2011), and is easily identifiable (Hibino et al., 2014). Previous work in Southern Japan has also ascertained that the taxon does not contain cryptic species (Reimer et al., 2007), making it a simple and reliable species for research and citizen science (Parkinson et al., 2016).

Specimens of P. tuberculosa were collected from January 2012 to November 2015 from eight locations around Okinawa-jima Island and one location on Amami Oshima Island, Kagoshima, to the north of Okinawa in the Middle Ryukyus (Table 1). All specimens were collected from the low intertidal zone (0–2 m depths depending on tides) via snorkeling. Small portions of colonies were collected and fixed in 70–99.5% ethanol for further molecular analyses. The collections were limited by the low numbers of P. tuberculosa present at some sites (n = 3–11).

Table 1 Sites from which P. tuberculosa specimens were collected in this study to examine Symbiodinium spp., and information on numbers of specimens, sea surface temperature (SST), and chlorophyll-a (chl-a) concentrations.

Site name	Latitude, longitude	# of specimens	SSTavg (°C)1	SSTmax ± SD (°C)2	SSTmin ± SD (°C)3	High August SST (year(s))	Low February SST (year(s))	Chl-a (mg/m3)4	
Wase (Amami)	28°17′37″N, 129°28′27″E	5	24.60	28.88 ± 0.93	20.35 ± 0.43	30.10 (2001, 2013)	19.50 (2009)	0.15	
Oku	26°50′53″N, 128°17′14″E	10	24.95	29.08 ± 0.60	20.69 ± 0.73	30.30 (2001)	18.70 (2008)	0.08	
Nerome	26°41′36″N, 128°6′28″E	8	24.85	29.33 ± 0.53	20.47 ± 0.59	30.45 (2001)	19.10 (2008)	0.30	
Bise	26°42′39″N, 127°52′52″E	11	24.95	29.21 ± 0.54	20.68 ± 0.46	30.30 (2001)	19.80 (2015)	0.15	
Teniya	26°33′51″N, 128°8′28″E	7	25.15	29.16 ± 0.57	20.82 ± 0.50	30.15 (2001)	20.25 (2002, 2009)	0.15	
Uken	26°22′46″N, 127°52′47″E	3	25.05	29.43 ± 0.76	20.39 ± 0.45	30.90 (2001)	19.55 (2011)	0.50	
Mizugama	26°21′35″N, 127°44′19″E	8	25.15	29.16 ± 0.51	21.21 ± 0.42	30.15 (2001)	20.70 (2015)	0.25	
Kyan	26°5′40″N, 127°39′10″E	4	25.35	29.18 ± 0.58	21.29 ± 0.62	30.15 (2001)	20.40 (2015)	0.24	
Odo	26°5′11″N, 127°42′37″E	7	25.35	29.23 ± 0.51	21.42 ± 0.60	30.15 (2001)	20.60 (2011)	0.08	
Notes:

1 Generated by Giovanni data (see Materials and Methods), average of all SST measurements taken May 2000–May 2015; value from generated map (standard deviation not available).

2 Average of highest SST observed in August each year (2000–2014).

3 Average of lowest SST observed in February each year (2000–2015).

4 Generated by Giovanni data (see Materials and Methods), average of all chl-a measurements taken July 2002–May 2015; value from generated map (standard deviation not available).

DNA extraction, PCR, and phylogenetic analyses

DNA was extracted from preserved colony samples using a DNeasy Blood and Tissue Kit (Qiagen, Tokyo, Japan) following the manufacturer’s protocol. We amplified two Symbiodinium DNA markers; the ITS2 in the ribosomal DNA array, and a portion of the plastid minicircle non-coding region (psbAncr). ITS2 sequences were obtained to place our new specimens within the phylogenetic framework of this well reported marker, and with past research on Symbiodinium within P. tuberculosa in Southern Japan (Reimer, Takishita & Maruyama, 2006), while psbAncr sequences were obtained to examine finer scale phylogenetic patterns (Reimer et al., 2017). ITS2 was amplified using the primers zITSf (5′-CCG GTG AAT TAT TCG GAC TGA CGC AGT-3′) and ITS4 (5′-TCC TCC GCT TAT TGA TAT GC-3′) (White et al., 1990; Rowan & Powers, 1992; Hunter, Morden & Smith, 1997). psbAncr was amplified using the primers 7.4-Forw (5′-GCA TGA AAG AAA TGC ACA CAA CTT CCC-3′) and 7.8-Rev (5′-GGT TCT CTT ATT CCA TCA ATA TCT ACT G-3′) (LaJeunesse & Thornhill, 2011). Reaction mixes contained 1.0 μl of genomic DNA, 7.0 μ1 of Milli-Q water, 10.0 μl of HotStarTaq Plus Master Mix, and 1.0 μl of each primer (10 pmol). Thermocycler conditions were as follows: for ITS2: 95.0 °C for 5 min; 35 cycles of 94.0 °C for 30 s, 51.0 °C for 45 s, and 72.0 °C for 2 min; 72.0 °C for 10 min; and for psbAncr: 95.0 °C for 5 min; 40 cycles of 94.0 °C for 10 s, 55.0 °C for 30 s, and 72.0 °C for 2 min; 72.0 °C for 10 min. Products were directly sequenced by Fasmac (Kanagawa, Japan). Novel sequences are deposited in GenBank under accession numbers MF582639–MF582657 and MF593399–MF593459.

The nucleotide sequences for ITS2 and psbAncr were separately aligned within Geneious v9.1.3 (Biomatters Limited, Auckland, New Zealand). Alignments were inspected manually, and primer regions and short sequences were excluded. Because the long plastid non-coding region rarely sequenced completely, we used only the forward psbAncr reads. The ITS2 alignment contained 24 sequences of 216 bp, while the psbAncr forward alignment contained 61 sequences of 300 bp. Previously reported sequences from GenBank were incorporated into the ITS2 alignment for reference (DQ480631, DQ480639, DQ889741, DQ889743—all Symbiodinium subclade C1 or closely related from P. tuberculosa from Southern Japan; and AB207184—Symbiodinium subclade C15 related from Zoanthus sp. in Southern Japan), while the psbAncr alignment contained only novel sequences (no previously reported sequences bore strong similarity).

Both alignments were analyzed using maximum likelihood (ML), neighbor-joining (NJ), maximum parsimony (MP) and Bayesian inference (BI) methods. ML analyses for both datasets were performed using PhyML (Guindon et al., 2010) with input trees generated by NJPlot (Perriere & Gouy, 1996) under automatic model selection by smart model selection with Akaike Information Criterion. Both datasets were analyzed under the HKY85 model (Hasegawa, Kishino & Yano, 1985) with the transition/transversion ratio estimated, the proportion of invariable sites fixed at 0.0, and the number of substitution rate categories as 1. PhyML bootstrap trees were made using the same parameters as the individual ML tree. The distances were calculated using a Kimura’s two-parameter model (Kimura, 1980). NJ analyses for both alignments were run within Geneious on default settings under the HKY85 model. MP analyses were performed in Paup* 4.0a147 (Swofford, 2000) with indels included as a fifth character state. All trees were run with 1,000 bootstraps. Bayesian posterior probabilities were calculated with the software Mr. Bayes (Huelsenbeck & Ronquist, 2001) using the HKY85 substitution model and default parameters (chain length = 1,000,000; burn-in = 250,000). Genetic distances between and within lineages were calculated in MEGA6 (Tamura et al., 2013) using the Maximum Composite Likelihood Model (Tamura, Nei & Kumar, 2004).

Results

Environmental data

Yearly average SST showed southern sites to be warmer than northern sites, with a difference of 0.95 °C between Wase on Amami Oshima Island (24.6 °C) compared to Kyan and Odo on the southern tip of Okinawa-jima Island (25.35 °C) (Table 1). For SSTmax, Wase was lowest (28.88 ± 0.93 °C) while Uken on the east coast of Okinawa-jima Island was highest (29.43 ± 0.76 °C) (Table 1). For SSTmin, Wase (20.35 ± 0.43 °C) was coldest, with highest SSTmin at Odo (21.42 ± 0.60 °C). The highest observed SST in any year was at Uken (30.9 °C in 2001), and the lowest was at Bise (18.7 °C in 2015) and Oku (18.7 °C in 2008) on the northwest and north coasts of Okinawa-jima Island, respectively (Table 1). Yearly average chl-a concentration values were generally low at all sampling sites, ranging from a low of 0.08 mg/m3 at Odo to a high of 0.50 mg/m3 at Uken (Table 1).

Phylogenetic analyses

Two different Symbiodinium ITS2 types were detected. The first type (n = 17) matched 100% with previously reported Symbiodinium subclade C1 from P. tuberculosa in Southern Japan (DQ889743; DQ889741) (Fig. 1A). The other type (n = 5) differed by one base pair and was also 100% identical to previously reported Symbiodinium from P. tuberculosa in Southern Japan (DQ480639). This second type formed a subclade within C1 (ML = 99%, NJ = 74%, MP = 61%, BI = 0.90) (Fig. 1A). These two types formed a large, moderately supported clade separate from subclade C3 (ML = 69%, NJ = 83%, MP = 64%, BI = 0.70), so we considered all of our sequences to be subclade C1 or “C1-related” (Fig. 1A; Table S1).

Figure 1 Phylogenies of clade C Symbiodinium isolated from Palythoa tuberculosa around Okinawa-jima Island and Amami Oshima Island, Japan.

Maximum likelihood (ML) trees are depicted for (A) the internal transcribed spacer 2 (ITS2) and (B) the chloroplast psbA noncoding region (psbAncr). Sequences from previous studies are included with GenBank accession numbers, host species, location, and subclade names sensu LaJeunesse (2001). Values at nodes represent ML, neighbor-joining (NJ), and maximum parsimony (MP) bootstrap percentages, as well as Bayesian inference (BI) posterior probabilities, respectively. Specimen abbreviations are as in Table S1.

Sixty-three unique Symbiodinium psbAncr forward sequences were recovered. Two sequences were too short to be included in the final alignment but were long enough to identify to lineage (described below, Table S1). The resulting psbAncr ML tree showed 33 specimens within a large, well-supported clade (ML = 100%, NJ = 100%, MP = 100%, BI = 1.00) that we designated as “lineage 1” (Fig. 1B). Specimens belonging to lineage 1 (n = 33) were recovered from all nine locations. Another 19 specimens from Oku, Teniya, and Mizugama formed a separate monophyly (“lineage 2”; ML = 76%, NJ = 93%, MP = 100%, BI = 0.98). Additionally, seven specimens from Wase, Nerome, Bise, Mizugama, and Odo formed a monophyly (“lineage 3”; ML = 75%, NJ = 100%, MP = 100%, BI = 0.77). Finally, two specimens from Bise and Nerome formed another monophyly (“lineage 4”; ML = 100%, NJ = 100%, MP = 100%, BI = 1.00). The between-lineage molecular distances for psbAncr ranged from 0.105 to 0.256, while the within-lineage distances were much smaller, ranging from 0.003 to 0.021 (Table 2).

Table 2 Pairwise genetic distances among Symbiodinium psbAncr lineages isolated from P. tuberculosa in Southern Japan.

	Lineage 1	Lineage 2	Lineage 3	Lineage 4*	
Lineage 1	0.003				
Lineage 2	0.137	0.005			
Lineage 3	0.184	0.128	0.021		
Lineage 4*	0.256	0.198	0.105	0.000*	
Notes:

Shaded diagonal values represent within-lineage distances.

* Lineage 4 was represented by two identical sequences.

Lineage distributions

We next examined the distribution of Symbiodinium lineages across locations (Fig. 2). Wase, Amami Oshima Island was dominated by lineage 1 (4/5 P. tuberculosa colonies), as was Bise (9/11), Uken (3/3), Kyan (4/4), and Odo (6/7). Lineage 2 was dominant at Oku (8/10), Teniya (5/7), and Mizugama (6/8), while lineage 3 was dominant at Nerome (5/8) and also appeared at Bise (2/7). Lineage 4 members only appeared in one colony each at Wase, Mizugama, and Odo.

Figure 2 Map of Amami Oshima Island (A) and Okinawa-jima Island (B) with average sea surface temperature (SSTavg) and Symbiodinium psbAncr lineage ratios at each site investigated.

Note thermal distortions near coastlines were ignored in all SST analyses as these are generated by influence of terrestrial portions of islands within the 4 km resolution of satellite data.

Finally, we examined the range of environments in which each lineage could be found (Fig. 3). Symbiodinium lineage 1 appeared at all sites and thus all environments in this study (SSTavg = 24.6–25.35 °C; SSTmax = 29.43 °C; SSTmin = 20.35 °C; chl-a 0.08–0.50 mg/m3). Lineage 2 was only observed at Oku, Teniya, and Mizugama, where SSTavg ranged between 24.95 and 25.15 °C, with SSTmin of 20.69 °C (Oku) and SSTmax of 29.16 °C (Teniya, Mizugama), and chl-a ranged between 0.08 and 0.25 mg/m3. Lineage 3 was only found at Bise and Nerome, with SSTavg of 24.85–24.95 °C, SSTmin of 20.47 °C, SSTmax of 29.33 °C (both Nerome), and chl-a of 0.15–0.30 mg/m3. Lineage 4 members were only observed once each at three locations, but these stretched across the geographic range of this study (Wase, Mizugama, Odo).

Figure 3 Symbiodinium psbAncr lineage distribution by environment.

Distributions are represented as (A) proportions of each lineage in each sampling site; or as ranges (dotted lines) with respect to (B) SSTavg values, (C) SSTmax values, (D) SSTmin values, and (E) chl-a values. All colors correspond to sample site designations in (A).

Discussion

Using the high-resolution psbAncr marker, we identify four Symbiodinium lineages associated with P. tuberculosa on Okinawa-jima Island. The lineages feature surprisingly unique distributions over a small geographic scale, and would be considered at most two entities based on lower-resolution ITS2 data (Fig. 1). Because the between-lineage molecular distances for psbAncr (0.105–0.256; Table 2) are greater than those reported between the psbAncr sequences of two divergent Symbiodinium ITS2 types (∼0.045 for C26a vs. C31; LaJeunesse & Thornhill, 2011), these lineages likely represent reproductively isolated species rather than populations within a species.

Lineage 1 appears to be a widely distributed generalist, at least over the range of this study in the Central Ryukyus (Fig. 2). It is found at all sites and often occupies the majority of colonies at a given site. Lineage 3 is observed only at the two sites on the northwestern coast of Okinawa-jima Island, and thus appears to have narrow geographical and environmental components to its distribution (SSTavg = 24.85–24.95 °C; chl-a = 0.15–0.30 mg/m3) (Fig. 3). Lineage 4 is present in very low numbers (n = 3) across the latitudinal range of the study, making it difficult to infer its environmental niche.

Lineage 2 has a somewhat restricted range, found at only three locations (where it also occupied the majority of colonies): Oku, Teniya, and Mizugama. These three sites are located near river outflows, suggesting this Symbiodinium lineage may be able to tolerate changes in salinity more effectively than the others. Unfortunately, fine-scale salinity data are not yet available to investigate this trend further. Chl-a levels at these sites are generally low (<0.25 mg/m3), while the SSTavg range is intermediate (24.95–25.15 °C). Lineage 2 is absent at other locations within this SSTavg range; for example, at Bise and Nerome, where only lineages 1 and 3 are present, and at Uken, where only lineage 1 is detected. Although Uken’s SSTavg is firmly in the middle of the SST range investigated in this study, Uken’s SSTmin and SSTmax are generally more extreme than those at other sites (Table 1) due to Uken’s position within shallow Southern Kin Bay, which is isolated from the stabilizing temperature effects of the open ocean (Montani, 1996). Additionally, chl-a levels at Uken are higher than those of all other locations (0.50 mg/m3).

By focusing on a very small area of the Northwest Pacific, and by using a rapidly evolving molecular marker, we could resolve a much finer scale of Symbiodinium biogeography than has previously been recognized in the region. Some psbAncr lineages appear to be partitioned on the basis of temperature (SSTmax) differences on the scale of 0.1–0.3 °C, much lower than the 0.7–1.0 °C observed in previous studies (Baums, Devlin-Durante & LaJeunesse, 2014; Reimer et al., 2017). It also appears SST stability (e.g., differences between SSTmax and SSTmin), as well as fine scale salinity differences (not measured here) may play an important role in the distribution of Symbiodinium diversity. Chl-a levels are generally low at all locations, although it should be noted that only generalist lineage 1 is found in Uken (although n = 3), the site with the highest chl-a levels. Overall, the contribution of chl-a (a proxy for turbidity) is not clear, as all P. tuberculosa specimens were collected from very shallow waters (0–2 m).

It is surprising that such biogeographic patterns could be uncovered given the low numbers of host P. tuberculosa at some locations. However, these low numbers also caution that the symbiont species’ distributions are unlikely to have been completely resolved. Another issue is that the fine-scale environmental variation among sites falls below the margin of error in the satellite datasets; future research of this nature must utilize more precise instrumentation. We did not examine host P. tuberculosa population genetics, and it remains to be seen if fine-scale host structure may also play a role in the observed patterns. Previous studies indicate cnidarian host and symbiont genetic structure can be mismatched (Baums, Devlin-Durante & LaJeunesse, 2014; Leydet & Hellberg, 2016), although clear cases of matching genetic structure have also been observed (Bongaerts et al., 2010; Prada et al., 2014).

As our ability to discern between different lineages of Symbiodinium has increased, so too has our understanding of the complexity and nuances of Symbiodinium diversity and distribution. The present work supports previous studies that show Symbiodinium evolution is driven largely by specialization to different environmental niches (Frade et al., 2008b; LaJeunesse et al., 2010; Kamezaki et al., 2013; Reimer et al., 2017), and that specialization may occur on much finer micro-environmental scales than usually addressed. Further characterization of the Symbiodinium–P. tuberculosa symbiosis within the Ryukyus or comparable island chains should be carried out to confirm similar symbiont structuring based on fine-scale environmental heterogeneity.

The Symbiodinium diversity patterns on Okinawa-jima Island highlight three major considerations for future investigations of this kind. First, it is advantageous for researchers to obtain fine-scale environmental data of the study area if available, so as to better characterize niches that might otherwise be deemed homogenous. Second, researchers should refrain from considering pooled specimens or results from different nearby locations as representative of a larger area without thoroughly checking for fine scale environmental patterns. Third, as has been suggested elsewhere (LaJeunesse & Thornhill, 2011; Reimer et al., 2017), researchers should incorporate both ITS2 data (to tie to past work) as well as rapidly evolving markers (to uncover hidden diversity) when investigating Symbiodinium biogeography, as a failure to address fine-scale niche adaptation could lead to a misinterpretation of results. These suggestions should improve the design of Symbiodinium studies on all geographic scales, from local to regional to global.

Supplemental Information

Supplemental Information 1 Table S1. Palythoa tuberculosa specimens examined in this study, their collection information, and Symbiodinium GenBank Accession Numbers.

Note not all specimens had their ITS2 sequences acquired as no clear distinctions asides from the two major groupings shown in Fig. 1A were found.

Click here for additional data file.

Supplemental Information 2 ITS alignment used in analyses.

Click here for additional data file.

Supplemental Information 3 psbA alignment used in analyses.

Click here for additional data file.

We would like to thank M. Mizuyama and I. Kawamura for research assistance, and O. Takama, S. Kunihiro, and M. Sakurai for help with specimen collection.

Additional Information and Declarations

Competing Interests

Author Contributions

DNA Deposition

Data Availability

James Reimer is an Academic Editor for PeerJ.

Hatsuko Noda conceived and designed the experiments, performed the experiments, analyzed the data, wrote the paper, prepared figures and/or tables, reviewed drafts of the paper.

John Everett Parkinson analyzed the data, prepared figures and/or tables, reviewed drafts of the paper.

Sung-Yin Yang conceived and designed the experiments, performed the experiments, prepared figures and/or tables.

James Davis Reimer conceived and designed the experiments, analyzed the data, contributed reagents/materials/analysis tools, wrote the paper, prepared figures and/or tables.

The following information was supplied regarding the deposition of DNA sequences:

GenBank: MF582639–MF582657 and MF593399–MF593459.

The following information was supplied regarding data availability:

The raw data has been provided as genetic data in the Supplementary Files, and specimens noted in tables.

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
