# Peer review of "A preliminary survey of zoantharian endosymbionts shows high genetic variation over small geographic scales on Okinawa-jima Island, Japan"

_PeerJ, doi:10.7717/peerj.3740_

## Round 0.1 · original submission · Minor Revisions

Both referees are supportive of this submission and provide only a few relatively straightforward suggestions for improvement. The second referee mentioned that they reviewed the manuscript previously and that all feedback from that has already been incorporated into the revised submission, so that they are now satisfied with this manuscript and compliment the authors on their revision. I expect that it will be a relatively quick and easy revision, and I look forward to seeing the revised manuscript.

Reviewer 1 ·

Basic reporting

1- Line 28: The verb “prefer” is not only too anthropomorphic, but also implies experimental manipulations were performed to investigate shifts in Symbiodinium communities. I strongly recommend that it be revised to “be associated with” as their study is purely one of associations between diversity and the environment.

2- Line 29: “slight” is awkward. Suggest “less than those usually investigated….”

3- Line 32: Suggest “accurately determine species diversity and their distributions.”

4- Intro, paragraph 2: Microsatellite markers have also been develop to investigate fine-scale diversity in Symbiodinium (see, Pettay & LaJeunesse 2007 [Molecular Ecology Notes], Grupstra et al. 2017 [Coral Reefs]).

5- Line 58: Suggest specifying “this work,” perhaps as “As a result of the development and implementation of finer resolution molecular markers…”

6- Line 65, for consistency, revise “reexamine” to “re-examine.”

7- Line 92: Another crucial aim was to explore potential associations between diversity and the environment, and is equally important to include at the end of the Introduction.

8- Line 122: Revise “Nucleic acids were extracted” to “DNA was extracted.”

9- Line 139: The psbA *gene* was not sequenced; the non-coding region was.

10- Line 142: Which previously reported sequences? Provide accession # for these here (as well as later in the Results as has been done).

11- Line 197: First sentence is awkward. Suggest, “We next examined the distribution of Symbiodinium lineages across locations.”

Experimental design

1- Line 86-87: Clarification is needed here (as well as later in the Methods for PCR and sequencing ITS2). If ITS2 sequences were already obtained for Symbiodinium within P. tuberculosa in southern Japan (*including Okinawa*), what was acquired in this study? New samples?

2- Line 124: According to TableS1, not all samples have a pending accession # for ITS2, implying that only a subset of samples were PCRed and sequenced for ITS2? Please clarify.

3- Line 131: Primer concentrations are needed.

4- Environmental data results: Were there any significant differences in temperature and chl-a between sites, for example, were southern sites *significantly* warmer than northern sites? A simple statistical test to test for significance between sites could add a great deal to explaining how/why the environment could shape the diversity observed. In contrast, if the sites are not significantly different, more caution should be made in declaring that the environment shapes diversity. Alternatively, address whether these environmental differences (whether significant or not) are biologically meaningful to the coral/symbiont relationship? Does a 0.95°C difference in temperature make a big difference in the coral/symbiont world?

Furthermore, “Yearly average chl-a concentration values were generally low” (Line 172 and later in Discussion), but no mention of whether there are apparent differences between sites to consider.

Finally, are there available software that could be used in this study to better test for diversity/environmental data associations? The program Geste and R ‘vegan’ package come to mind in using generalized linear modeling to investigate whether genetic structure relates to environmental factors. However, I am unsure whether the sample sizes in this study are sufficient or whether sequence data are able to be analyzed by these programs. Worth looking into these and perhaps other options.

5- Why the ITS2 marker was used at all needs additional justification (in addition to Line 277- the end of the paper). Its results are hardly ever mentioned/discussed in the discussion and understandably so; however, it needs to be made more clear why this marker was used in the first place (in Methods), and why the results from psbA noncoding region were the focus of the conclusions and discussion.

Validity of the findings

Overall, the findings and conclusions drawn are sound. I appreciate their recognition of the limitations of their study Lines 252-260. As pointed out in this paragraph, host variation could explain Symbiodinium diversity. While they cite cases of mismatch between host and symbiont, they should also reference studies that have found a match suggesting Symbiodinium diversity/distribution is simply due to host-specificity (examples include Bongaerts et al. 2010 [PLoS One], Prada et al. 2014 [Mol Eco]).

Additional comments

Noda et al. use a faster evolving gene to elucidate the diversity of Symbiodinium within Palythoa tuberculosa in Japan, and combine it with fine-scale environmental data to explore any associations between this diversity and the environment. This work is crucial to the field in joining a few other recent papers that advocate for such a finer-scale approach to better understand the biogeography of Symbiodinium. Overall, the study is sound and robust. The paper is overall well written and this review presents minor comments and revision suggestions.

Reviewer 2 ·

Basic reporting

The authors present a clear and unambiguous analysis of Symbiodinium diversity in Palythoa tuberculosis using a highly resolving genetic marker. Using satellite derived environmental data, the authors discuss the fine-scale zonation of different zoox haplotypes and discuss this in the context of the importance of fine-scale surveys to accurately capture Symbiodinium diversity in corals. I am glad to see more manuscripts moving away from the classical single-marker ITS2 approach and towards a multiple-marker approach. Based on the improvements made to this version of the manuscript (I reviewed a previous version), I am happy to recommend this manuscript for publication in PeerJ. The next obvious question that arrises from this work is what are the functional differences between psba haplotypes?

Experimental design

no comment

Validity of the findings

no comment

---

## Round 0.2 · accepted · Accept

I am satisfied with the revisions and responses to the referee comments, and am happy to move this manuscript forward into production.